# Use of a Dielectric Sensor for Salinity Determination on an Extensive Green Roof Substrate

**DOI:** 10.3390/s23135802

**Published:** 2023-06-21

**Authors:** Georgios Kargas, Nikolaos Ntoulas, Andreas Tsapatsouli

**Affiliations:** 1Laboratory of Agricultural Hydraulic, Division of Water Resources Management, Department of Natural Resources Management and Agricultural Engineering, Agricultural University of Athens, Iera Odos 75, 11855 Athens, Greece; kargas@aua.gr; 2Laboratory of Floriculture and Landscape Architecture, Department of Crop Science, Agricultural University of Athens, Iera Odos 75, 11855 Athens, Greece; andreas_tsapa@hotmail.com

**Keywords:** bulk electrical conductivity, dielectric permittivity, leachate electrical conductivity, salinity index, turfgrass

## Abstract

The irrigation of extensive green roofs with recycled or saline water could contribute to the conservation of valuable drinking water supplies. In such cases, the continuous monitoring of substrate electrical conductivity (EC_sw_) is of immense importance for the sustainable growth of the plants growing on the green roof. The present study aimed to estimate the EC_sw_ (pore water EC) of an extensive green roof substrate in lysimeters with the use of the WET-2 dielectric sensor. Half of the 48 lysimeters that simulated extensive green roofs had a substrate depth of 7.5 cm, while the other half had a 15 cm substrate depth. The warm season turfgrass *Paspalum vaginatum* ‘Platinum TE’ was established at the lysimeters, and during the summer period, it was irrigated every two days at a rate of 14 mm with NaCl solutions of various electrical conductivities (EC_i_): (a) 3 dS m^−1^, (b) 6 dS m^−1^, and (c) 12 dS m^−1^, while potable water of 0.3 dS m^−1^ EC_i_ served as the control. The relation between bulk electrical conductivity, σ_b_, and bulk dielectric permittivity, ε_b_, of the substrate was observed to be linear for all EC_i_ levels up to σ_b_ values of 2–2.5 dS m^−1^. The EC_sw_ was predicted by employing the salinity index method which was modified to be applied to the particular case of a green roof substrate. Knowing the salinity index and organic portion (%, *v*/*v*) for a given green roof substrate, we could calculate the EC_sw_. It was found that the use of the salinity index method predicts reliably the EC_sw_ up to 10–11 dS m^−1^, while the method overestimates EC_sw_ at very low levels of electrical conductivity.

## 1. Introduction

In recent decades, green roofs are considered a common practice for reintroducing open green spaces in the densely built fabric of contemporary large cities. Green roofs, regardless of their type, comprise a multi-layered system that commonly includes a protection mat for the waterproofing membrane, a drainage layer, a filtering sheet, a plant growth substrate comprised of lightweight materials, and vegetation [1]. According to existing guidelines [2], green roofs with a substrate depth of up to 15 cm are classified as extensive types, and their vegetation consists of low-maintenance and drought-resistant plants.

Numerous requirements must be met by the growing substrate for extensive green roofs, including maintaining the necessary substrate moisture for plant growth, enabling the quick removal of excess water, providing support and anchoring for the plants, providing nutrients, and having a pH and substrate electrical conductivity (EC_sw_) that are suitable for plant growth [3]. The constituents of extensive green roof substrates are mainly inorganic [4], lightweight, and coarse, and possess significant internal porosity, such as pumice [5], crushed bricks or tiles [6], zeolite [7], heat-expanded shale, clay or slate [8], perlite [9], and lava [10]. Organic substances, such as peat and composts, are also used to improve the physical and chemical properties of the substrate, but at smaller participation percentages to prevent substrate subsidizing as a result of decomposition [11].

The benefits of green roofs are numerous, including the amelioration of the urban heat-island effect [12], a reduction in air pollution [13], and stormwater management [14]. However, the implementation of green roofs on a large scale is required to significantly improve urban microclimates [15]. As a result, to meet this increased demand for the irrigation of green roofs, especially in arid or semi-arid areas, such as the Mediterranean region, alternative water irrigation sources should be explored. Saline groundwater, brackish surface water, grey water, or recycled water could all be used for green roof irrigation [16,17]. A significant disadvantage of the aforementioned alternative water sources is the increased electrical conductivity (EC). Hence, to ensure the long-term viability of the plants growing on green roofs, the EC_sw_ must be precisely determined and continuously monitored.

The salinity in a soil environment is mainly estimated by measuring the EC of the saturated soil paste extract (EC_e_) [18], which is now established as a standard method. However, this method is laborious and time-consuming since it involves soil sampling, the creation of saturated paste, the collection of extract, and the measurement of the EC_e_ [19]. As a result, instead of EC_e_, in many cases, electrical conductivity is determined using soil–water solutions in various ratios. This method of indirect estimation of EC_e_ is practically easier but requires that the relationship between EC_e_ and the EC of the specific soil/water ratio be known in advance to be able to estimate EC_e_.

Dielectric devices, such as time domain reflectometry (TDR) sensors and frequency domain reflectometry (FDR) sensors, have made it possible to simultaneously measure the bulk dielectric permittivity (ε_b_), the bulk electrical conductivity (σ_b_), and the temperature at the same point in the soil. From the measurement of σ_b_, the electrical conductivity of the pore water solution of the soil or growing substrate (EC_sw_) can be calculated with the help of various models [20,21,22,23,24,25,26].

Malicki and Walczak [21] introduced the concept of the salinity index (X_s_), utilizing measurements of ε_b_ and σ_b_ obtained by the TDR device. Then, Wilczak et al. [26] used the salinity index method to calculate EC_sw_ from ε_b_ and σ_b_ data acquired by an FDR sensor operating at a frequency of up to 500 MHz.

The salinity index (X_s_) is defined as the partial derivative of σ_b_ to ε_b_, where both parameters are determined by a dielectric sensor at the same time and the same point in the soil.
(1)$$X_{s}=\frac{\partial \sigma_{b}}{\partial \varepsilon_{b}}$$

Additionally, Malicli and Walczak [21] reported that the X_s_ is independent of soil water content (θ) when it is above 0.2 cm^3^ cm^−3^ and that X_s_ depends on soil salinity and texture. The relationship between σ_b_ and ε_b_ was proved to be linear, and the slope, which equals the X_s_, increases with the increasing EC of the irrigation water (EC_i_) in the same porous medium but is different for the same value of EC_i_ in different porous media. The constant term (y-intercept) of the linear relationship also depends on the type of medium and the value of EC_i_.

When the value of X_s_ and the slope (l) of the relationship X_s_–EC_i_ are known, then it is possible to calculate the value of EC_sw_ from the equation [21]:(2)$$EC_{sw}=\frac{X_{s}}{l}$$

Malicki and Walczak [21], for the case of inorganic soils, proposed an equation (Equation (3)) to calculate X_s_ (for values of σ_b_ > 0.08 dS m^−1^ and ε_b_ > 6.2).
(3)$$X_{s}=\frac{\sigma_{b}-0.08}{\varepsilon_{b}-6.2}$$
where σ_b_ = 0.08 dS m^−1^ and ε_b_ = 6.2 is the common point where the lines converge σ_b_ (ε_b_, EC_sw_). 

They also showed that slope l can be calculated from an empirical relationship based on the sand content of the soil. The final suggested EC_sw_ calculation equation is:(4)$$EC_{sw}=\frac{Xs}{l}=\frac{\sigma_{b}-0.08}{\left(\varepsilon_{b}-6.2\right)\left(0.0057+0.00007S\right)}$$
where S is the % (*w*/*w*) sand content of the soil. Equation (4) is valid for values of θ langer than 0.2 cm^3^ cm^−3^ because, as mentioned by Malicki and Walczak [21], above this limit, X_s_ is independent of the θ value.

The existence of a linear relationship between σ_b_ and ε_b_ from data obtained from the WET-2 sensor (Delta-T Devices, Cambridge, UK), which operates at a frequency of 20 MHz, has been confirmed in many cases of inorganic porous media [23,24,27]. However, the relationship between σ_b_ and ε_b_ has not been investigated in depth until now for coarse textured growing substrates suitable for extensive green roofs [28], as well as the possibility of using the empirical model of Malicki and Walczak [21] (Equation (4)) for EC_sw_ prediction on such substrates. However, to implement this model, it is necessary to know the point of intersection of the linear relationships σ_b_–ε_b,_ as in inorganic soils, but also a corresponding correlation of *l* with the substrate properties. However, in the case of green roof substrates, there is zero sand content, so Equation (4) cannot be used. In these cases, an empirical relationship must be found that relates l to another property of the substrate, for instance, the volumetric portion of clay, pumice, perlite, or of organic substances. To be able to find such a relationship requires much more experimental data from different substrates. Such relations for the case of soils have been presented by Wilczak et al. [26] and Kargas and Kerkides [29].

The primary aim of this investigation was to evaluate the EC_sw_ of a green roof substrate, at two different substrate depths, when irrigation was applied with water of EC_i_ of 0.3 dS m^−1^, 3 dS m^−1^, 6 dS m^−1^, and 12 dS m^−1^, employing a modified salinity index method suitable for extensive green roof substrates. Specifically, the objectives of this study were, (a) to verify the linear relationship between σ_b_–ε_b_ for ΕC_i_ values, (b) to establish, if there is a limit to the value of σ_b_, above which the relationship σ_b_–ε_b_ is not linear, and (c) to find an empirical relationship that relates *l* with the organic portion (%, *v*/*v*) of the green roof substrate.

## 2. Materials and Methods

### 2.1. Experimental Setup

The outdoor study was conducted at the experimental field of the Laboratory of Floriculture and Landscape Architecture, Agricultural University of Athens, Greece (37°59′ N, 23°42′ E, 35 m above sea level) from 19 July 2014 until 23 October 2014. It comprised 48 PVC lysimeters with a 300 mm inner diameter placed on leveled benches. Within each lysimeter, a complete layered simulation of an extensive green roof system was constructed. More specifically, the bottom of the lysimeter was covered with a protective mat made of non-rotting synthetic polyester fibers with a 3 mm thickness, a dry weight of 0.32 kg m^−2^, and a capacity to retain 3 L m^−2^ of water (TSM32, Zinco, Egreen, Athens, Greece). A drainage board layer made of recycled polyethylene with a height of 25 mm and a weight of 1.5 kg m^−2^ (FD25, Zinco, Egreen, Athens, Greece) was placed on top of the protective cloth. The drainage layer, which was equipped with water-retaining troughs, could store 3 L m^−2^. The drainage layer was covered with a non-woven geotextile (SF, Zinco, Egreen, Athens, Greece) made of thermally strengthened polypropylene, with a thickness of 600 μm, a weight of 100 g m^−2^, a 90% apparent opening size of 95 mm, and a water flow rate of 0.07 m s^−1^. A 10 mm diameter outflow opening was constructed at the bottom center of each lysimeter. The leachate was directed into a 2 L tank which was kept beneath the lysimeter by a flexible hose that was connected to the outlet.

The lysimeters were filled with a specialized and patented green roof substrate (Patent No. 1008610), which comprised 65% pumice, 15% thermally treated attapulgite clay, 15% grape marc compost, and 5% clinoptilolite zeolite by volume. The physical and chemical properties of the substrate are presented in Table 1. The substrate depth in half of the lysimeters was 7.5 cm, whereas, in the other half, it was 15 cm. After substrates were placed into the lysimeters, light compression and leveling were applied. 

### 2.2. Turfgrass Establishment and Irrigation

The warm season turfgrass species *Paspalum vaginatum* ‘Platinum TE’ was established in the plots 45 days before the initiation of the study on 2 June 2014 using washed sod. *Paspalum vaginatum* (seashore paspalum) was selected because it exhibits several desirable characteristics including increased tolerance to irrigation with water of high salinity up to seawater levels [8,30].

After sodding, the lysimeters were irrigated daily with potable water (EC of 0.3 dS m^−1^) until the initiation of increased salinity irrigation treatments. Increased salinity treatments were initiated on 19 July and ended on 23 October 2014, totaling 97 d. At the initiation of the study (17 July 2014), all lysimeters were irrigated with ample potable water (EC of 0.3 dS m^−1^) to produce uniform substrate moisture conditions. From then on, seashore paspalum was irrigated every other day at the depth of 14 mm (7 mm per day) with three NaCl irrigation solutions (EC_i_) of 3 dS m^−1^, 6 dS m^−1^, and 12 dS m^−1^, while potable water of 0.3 dS m^−1^ electrical conductivity served as the control. 

The lysimeters were hand-irrigated with a watering can with a nozzle to ensure even water distribution. During the first three irrigation events of the study (19, 21, and 23 July 2014), irrigation solution of 3 dS m^−1^ was used for all salinity treatments to avoid salinity shock of the turfgrass. From then on, irrigation was applied according to the experimental setup with all three salinity solutions (3 dS m^−1^, 6 dS m^−1^, and 12 dS m^−1^).

### 2.3. Meteorological Data

During the study period (17 July 2014 until 23 October 2014), air temperature and precipitation were monitored by the weather station of the Laboratory of General and Agricultural Meteorology of the Agricultural University of Athens, Athens, Greece, located 15 m away from the experimental site (Figure 1). No precipitation was observed during the summer period of the study (17 July–31 August 2014), having an average air temperature of 27.7 °C. The first autumn rainfall was recorded on 9 September 2014 (11.8 mm). Rainfall was also recorded on 16 September (1.8 mm), 7 October (4.6 mm), 22 October (4 mm), and 23 October (26.6 mm), marking the end of the study. The mean air temperature during the autumn months of the study (1 September–23 October 2014) was 22.0 °C.

### 2.4. Measurements

Throughout the study, the volume and salinity of the leachate, which was collected in tanks beneath each lysimeter, were determined every two days just before irrigation events. The collected leachate in the tanks referred to the previous irrigation application and was discarded after the determination of its volume and EC. For the electrical conductivity of the leachate (EC_L_) measurements, a handheld conductivity meter (CyberScan 200, Eutech Instruments Pte Ltd., Singapore) was used, which automatically corrected the EC to 25 °C. The outflow runoff volume from each lysimeter was divided by the inflow irrigation water volume to estimate the leaching fraction (LF).

Besides EC_L_ measurements, the substrate bulk dielectric permittivity (ε_b_) as well as the bulk electrical conductivity of the substrate (σ_b_) were determined. The determination was made every four days, before irrigation, using the WET-2 sensor (Delta-T Devices, Cambridge, UK). The WET-2 sensor is an inexpensive frequency domain dielectric soil moisture sensor, which is not able to automatically record data, and its probe consists of three metal rods. The rods are 6.8 cm long and spaced 1.5 cm apart, creating a cylindrical sampling area 65 mm deep and 45 mm wide [31]. The sensor was connected to an HH2 portable moisture meter (Delta-T Devices), in which the Hilhorst [22] model (Equation (5)) is installed for EC_sw_ determinations.
(5)$$EC_{sw}=\frac{80\sigma_{b}}{\varepsilon_{b}-4.1}$$

### 2.5. Experimental Methodology and Statistics

The experimental design was multi-factorial and involved two factors: two green roof substrate depths (7.5 cm or 15 cm) and four irrigation water salinities (0.3 dS m^−1^, 3 dS m^−1^, 6 dS m^−1^, and 12 dS m^−1^). The plot arrangement followed a completely randomized design, and each treatment was replicated 6 times resulting in 48 lysimeters (4 irrigation treatments × 2 substrate depths × 6 replications = 48 lysimeters).

In the initial stage of the current study, regressions were conducted between σ_b_ and ε_b_, which were determined using the WET-2 sensor. These regressions were performed for both substrate depths and each level of EC_i_ using the JMP® ver.11 statistical software (SAS Institute Inc., Cary, NC, USA). Based on the outcomes of these regressions, the viability of utilizing Equations (3) and (4) proposed by Malicki and Walczak [21] for X_s_ and EC_sw_ value calculation was assessed in the context of a green roof substrate. 

Subsequently, the feasibility of employing Equation (2), using the value of X_s_ and the slope (l) of the relationship X_s_–EC_i_ to estimate the EC_sw_ value of the green roof substrate utilized in the study was examined. This investigation was carried out for each value of EC_i_, for both substrate depths. The calculated average EC_sw_ value was then compared to the measured average EC_L_ value for each level of EC_i_. Considering the shallow substrate depths, substrate characteristics, high EC_i_ values, and the uniform development of the root system, it is reasonable to argue that the actual salinity of the substrate closely resembles the salinity of the leachate. These factors contribute to a more homogenous distribution of salinity within the substrate, minimizing potential variations and ensuring that the leachate represents the substrate’s salinity levels.

In conclusion, a modification of the Malicki and Walzczak [21] model for green roof substrates is presented. This modification is based on the proposed empirical relationship linking *l* with the organic portion (%, *v*/*v*) of the green roof substrate, as well as the new version of the salinity index (Xs) for green roof substrates. To assess the accuracy of the modified Malicki and Walczak model in predicting EC_sw_ for green roof substrates, the root-mean-square error (RMSE) was utilized. By comparing the predicted EC_sw_ values with the measured EC_L_ values, the effectiveness of the model was evaluated.

## 3. Results and Discussion

The electrical conductivity of the leachate collected in tanks beneath the lysimeters started to increase immediately after the initiation of high-salinity irrigation treatments (Figure 2). When irrigation was applied with the 3 dS m^−1^ solution, the equalization of EC_L_ with that of the EC_i_ was recorded 10 days after initiation (DAI) of the study for the 7.5 cm substrate depth and 16 DAI for the 15 cm substrate depth. Accordingly, when irrigation was applied to 6 dS m^−1^ and 12 dS m^−1^ solutions, the equalization of EC_L_ with EC_i_ occurred on 16 and 20 DAI for 7.5 cm and 15 cm substrate depth, respectively, since an irrigation solution of 3 dS m^−1^ was used for all salinity treatments during the study’s first three irrigation events to prevent the salinity shock of the turfgrass. During August, successive irrigation events gradually increased EC_L_, reaching values of 13–14 dS m^−1^ for EC_i_ = 3 dS m^−1^, 20–21 dS m^−1^ for EC_i_ = 6 dS m^−1^, and 31–32 dS m^−1^ for EC_i_ = 12 dS m^−1^. In contrast, the EC_L_ values of the experimental plots irrigated exclusively with potable water remained relatively constant throughout the study at relatively low levels (<1.5 dS m^−1^), above the potable water’s EC value (0.3 dS m^−1^). This sharp increase in EC_L_ is attributed to the very low LF which favored the continuous accumulation of salts at the substrate and the drainage layers of the simulated green roofs in the lysimeters. More specifically, from the beginning of the study until the end of August, the LF had an average value for all irrigation treatments less than 0.1 (≈0.09), which even prevented EC_L_ determination during several sampling periods (18–22 August and 28 August–1 September). The reduction in temperatures throughout September and the presence of light rains (Figure 1) favored an increase in LF, with a mean value of 0.55, resulting in an EC_L_ gradual decrease. The increase in LF continued during October with a mean value of 0.75, promoting further EC_L_ reduction, which by the end of the study (23 October 2014), reached values of approximately 5 dS m^−1^ for EC_i_ = 3 dS m^−1^, 8.5 dS m^−1^ for EC_i_ = 6 dS m^−1^, and 16.5 dS m^−1^ for EC_i_ = 12 dS m^−1^, for both substrate depths. It is apparent that whenever green roof irrigation is applied using high-salinity water, leaching requirements are expected to be high enough for EC_L_ to remain close to the EC_i_ [8].

An important observation from the study was that the green coverage of the *P. vaginatum* turfgrass consistently remained high, exceeding 90% across all treatments. This finding demonstrates the remarkable resilience of seashore paspalum to elevated salinity levels [32,33] and highlights the potential for irrigating *P. vaginatum* turfgrass in extensive green roofs with water with an EC of up to 12 dS m^−1^ for extended periods without a significant deterioration in visual quality.

According to Figure 3, the relationship between σ_b_ and ε_b_ is strongly linear for all levels of EC_i_ for both substrate depths of 7.5 cm and 15 cm, and the correlation coefficient R^2^ decreases with increasing EC_i_. For the substrate depth of 7.5 cm the values of R^2^ are equal to 0.84, 0.90, 0.77, and 0.57 for irrigation with potable water and solutions with EC_i_ of 3 dS m^−1^, 6 dS m^−1^, and 12 dS m^−1^, respectively, while for the depth of 15 cm, they are equal to 0.74, 0.87, 0.80, and 0.70, respectively. The large decrease in the R^2^ value for EC_i_ = 12 dS m^−1^, when the substrate depth was 7.5 cm, is probably because several values of σ_b_ were greater than 3 dS m^−1^. According to Kargas et al. [34], the values of ε_b_ from the WET-2 sensor for soils are reliable up to about the value σ_b_ = 3 dS m^−1^ since for higher values of σ_b_, there is a decrease in the value of ε_b_. As can be observed from the results of this present study, for a reliable estimate of ε_b_ in the case of green roof substrates, it is better to limit σ_b_ up to about values of 2–2.5 dS m^−1^. Additionally, from Figure 3, the point of intersection of the relations σ_b_–ε_b_ is quite different from that of inorganic soils, so the proposed Equation (3) of Malicki and Walczak [21] for X_s_ value calculation cannot be used in the case of green roof substrates. Roughly, it can be seen that the intersection point has values of ε_b_ = 12 and σ_b_ = 0.125 dS m^−1^, which are very different from the intersection points of inorganic soils, ε_b_ =6.2 and σ_b_ = 0.08 dS m^−1^.

Observing the slopes of the σ_b_–ε_b_ relationships, which are equal to the value of the salinity index (X_s_), for both substrate depths, they increase with the increase in the EC_i_ values and are therefore proportional to the salinity regime established in the substrate. More specifically, the values of X_s_ vary from 0.0593 to 0.0109 for the depth of 7.5 cm and from 0.0544 to 0.0109 for the depth of 15 cm, depending on the value of EC_i_ (Figure 3). Moreover, the value of the constant term of the linear relationships decreases with increasing EC_i_ value. Thus, for EC_i_ = 0.3 dS m^−1^, the constant term has a value of −0.0570, while for EC_i_ = 12 dS m^−1^, it has a value of −0.2749 for the depth of 7.5 cm, whereas, for the depth of 15 cm for EC_i_ = 0.3 dS m^−1^, the constant term has a value of −0.0409, while for EC_i_ = 12 dS m^−1^, it has a value of −0.4425.

In Figure 4, the relationship between X_s_ and EC_i_ for the two substrate depths of 7.5 cm and 15 cm is presented. These relationships are strongly linear with R^2^ = 0.90 and a slope of 0.0040 for a depth of 7.5 cm and R^2^ = 0.95 and a slope of 0.0035 for a depth of 15 cm. The slope value for the green roof substrate used in the study at both depths is completely different (lower) from those presented by Malicki and Walczak [21] for various soil types. For sandy soil, they reported values of 0.0136 and 0.0126; for loamy sand soil, a value of 0.011; for silty loam soil, a value of 0.0098; and for silt, a value of 0.0081. Therefore, the X_s_–EC_i_ relationship, like the point of intersection of the σ_b_–ε_b_ lines, depends on the special characteristics of the porous medium. Due to the aforementioned information, Equation (4) proposed above for calculating the EC_sw_ cannot be used in the case of green roof substrates because they have completely different characteristics. As a first approach, Equation (2) could be used to estimate the EC_sw_ value, which will be built up in the substrate for each value of EC_i_.

For the substrate depth of 7.5 cm, the estimated mean EC_sw_, using Equation (2) for the corresponding salinity indices, is 2.73 dS m^−1^ for irrigation with potable water, 7.80 dS m^−1^ for EC_i_ = 3 dS m^−1^, 12.08 dS m^−1^ for EC_i_ = 6 dS m^−1^, and 14.83 dS m^−1^ for EC_i_ = 12 dS m^−1^, while for the 15 cm substrate depth, the estimated mean EC_sw_ is 3.11 dS m^−1^ for irrigation with potable water, 8.57 dS m^−1^ for EC_i_ = 3 dS m^−1^, 9.83 dS m^−1^ for EC_i_ = 6 dS m^−1^, and 15.54 dS m^−1^ for EC_i_ = 12 dS m^−1^ (Table 2). For both substrate depths, from the measurement of mean EC_L_ for the study period, it can be claimed that the salinity index model accurately predicts the EC_sw_ for irrigation with an EC_i_ of 3 dS m^−1^ and 6 dS m^−1^, while it significantly overestimates for small values of the salinity index such as 0.0109, which corresponds to potable water (EC_i_ = 0.3 dS m^−1^) (Table 2). For those cases where irrigation is applied with low EC_i_ values, EC_sw_ can be predicted with the Hilhorst [22] model (Equation (5)), which is installed into the HH2 meter, to which the WET-2 sensor is connected. The mean value of EC_sw_ for the study period for irrigation with potable water as measured with the WET-2 sensor and using the Hilhorst model is 0.67 dS m^−1^ for the 7.5 cm substrate depth and 0.85 dS m^−1^ for the 15 cm substrate depth, which is close to the measured mean values of the EC_L_ equal to 0.53 dS m^−1^ and 0.78 dS m^−1^ for the substrate depths of 7.5 cm and 15 cm, respectively. On the contrary, the Hilhorst model underestimated the EC_sw_ when irrigation was applied to high salinity solutions of 3, 6, and 12 dS m^−1^. More specifically, the mean EC_sw_ value measured with the Hilhorst model is equal to 2.18 dS m^−1^, 2.97 dS m^−1^, and 4.49 dS m^−1^ for the substrate depth of 7.5 cm and 2.07 dS m^−1^, 2.64 dS m^−1^, and 3.80 dS m^−1^ for the substrate depth of 15 cm for EC_i_ of 3 dS m^−1^, 6 dS m^−1^, and 12 dS m^−1^, respectively, when the measured mean values of the EC_L_ equal to 6.86 dS m^−1^, 11.14 dS m^−1^, and 19.73 dS m^−1^ for the substrate depth of 7.5 cm and 7.51 dS m^−1^, 10.54 dS m^−1^, and 18.56 dS m^−1^ for the substrate depth of 15 cm, respectively (Table 2). Similar behavior of the Hilhorst model was also observed by Bañón et al. [28] in measurements with the dielectric soil moisture sensor GS3 on a substrate comprised of 60% sphagnum peat, 30% coconut fiber, and 10% perlite by volume. In particular, the deviations (underestimation) increased when the substrate water content was very low and salinity was high.

For irrigation with EC_i_ = 12 dS m^−1^, the salinity index model significantly underestimates EC_sw_ for both substrate depths by calculating an EC_sw_ value equal to 14.83 dS m^−1^ for the 7.5 cm substrate depth and 15.54 dS m^−1^ for the 15 cm substrate depth, while the measured mean values of the EC_L_ were equal to 19.73 dS m^−1^ and 18.56 dS m^−1^ for the substrate depth of 7.5 cm and 15 cm, respectively. This can be attributed to the fact that several values of σ_b_ were greater than 3 dS m^−1^ when irrigation was applied with EC_i_ = 12 dS m^−1^ resulting in false ε_b_ values (Figure 3) [34]. This is also confirmed by the low R^2^ values (0.57 for the 7.5 cm substrate depth and 0.70 for the 15 cm substrate depth) of the σ_b_–ε_b_ relationships, and, as a consequence, the slope of these regression lines, which equals the X_s_, is unreliable.

To calculate the value of X_s_ using the method mentioned above, it is necessary to obtain a series of measurements on the same substrate with the same solution at different θ. Obviously, this procedure has little practical value in the calculation of EC_sw_. However, if the point of convergence of the lines is known then it is possible to calculate the value of X_s_ by determining the value of the partial derivative from a suitable difference quotient. Considering that the convergence point has values of ε_b_ = 12 and σ_b_ = 0.145 dS m^−1^, the value of X_s_ can be approximated by the equation:(6)$$X_{s}=\frac{\sigma_{b}-0.145}{\varepsilon_{b}-12}$$

However, to make the model more functional in terms of EC_sw_ estimation, the slope l (Equation (2)) must be related to some property of the green roof substrates, e.g., the volumetric portion of pumice or organic substance, as was performed with the classic Malicki and Walczak model [21], where the slope l of soils was correlated with sand content. For this, similar data are needed from other types of substrates in order to draw safe conclusions [29]. From the literature review, it was found that experimental data from different growing substrates are extremely limited. In a study by Incrocci et al. [35], experimental data of a substrate comprising peat and pumice (1:1, *v*/*v*) are presented. After an appropriate transformation of the data, it follows that the relationship σ_b_–ε_b_ is linear at each EC_i_ value, just like the experimental data in the present study, but also that the relationship between X_s_ and EC_i_ is linear with a slope of 0.0123. If the slope l is related to the organic portion (% *v*/*v*) of the two substrates, peat/pumice (1:1, *v*:*v*) and the substrate evaluated in the present study comprised pumice/thermally treated attapulgite clay/clinoptilolite zeolite/grape marc compost (65:15:5:15, *v*:*v*), then the relationship is:(7)l=0.0002OM+0.0012
where OM is the organic portion (% *v*/*v*) of the substrate.

Of course, it is important to note that while these results are promising, they are only limited to two growing substrates. It is, therefore, necessary to extend this investigation, also considering other growing substrates which are widely used in horticulture and green roofs.

Thus, from the experimental data up to now, in the case of the two substrates, the model of Malicki and Walczak [21] can acquire the particular form:(8)$$EC_{sw}=\frac{Xs}{l}=\frac{\sigma_{b}-0.145}{\left(\varepsilon_{b}-12\right)\left(0.0012+0.0002OM\right)}$$

Therefore, from the measurement of σ_b_ and ε_b_ and the organic portion (% *v*/*v*) of the substrate, the value of EC_sw_ can be estimated. Based on the estimated EC_sw_ and EC_L_ measurements during the study, the root mean square error (RSME) was calculated equal to 1.10, 1.69, 2.08, and 4.29 for irrigation with potable water and solutions with EC_i_ of 3 dS m^−1^, 6 dS m^−1^, and 12 dS m^−1^, respectively, when the substrate depth was 7.5 cm, while for the depth of 15 cm, they were equal to 0.96, 1.47, 2.39, and 4.62, respectively. These results indicate that Equation (8) provided adequately reliable results when irrigation was applied with water of 3 dS m^−1^ and 6 dS m^−1^ EC_i_. When irrigation is applied with low EC_i_ values such as those of potable water, the Hilhorst [22] model (Equation (5)), which is installed in the HH2 meter and to which the WET-2 sensor is connected, can be used to predict EC_sw_.

## 4. Conclusions

Based on the results of the study, it was found that the relationship σ_b_–ε_b_ for an extensive green roof substrate is linear for each level of EC_i_, and the slope of the linear relationship increases with the increase in EC_i_. The linearity is maintained up to a level of σ_b_ values, which depends on the characteristics of the dielectric sensor.

However, the empirical relationship presented by Malicki and Walczak [21] for in-organic soils cannot be used to predict EC_sw_ of extensive green roofs because the relationships for calculating the salinity index and the slope l in the green roof substrates differ significantly. Applying the Malicki and Walczak [21] model to the general form showed that the model generally accurately predicts the EC_sw_ of extensive green roofs for EC_sw_ up to 10–11 dS m^−1^ while significantly overestimating at very low salinity index values.

With the appropriate modification of the Malicki and Walczak model for the case of green roof substrates, to predict EC_sw_, the determination of the WET-based salinity index (X_s_) and organic portion (% *v*/*v*) of the substrate is required. The modified Malicki and Walczak model is more effective than the Hilhorst model in all cases except for the cases when the EC_sw_ is lower than 2 dS m^−1^.

## Figures and Tables

**Figure 1 sensors-23-05802-f001:**
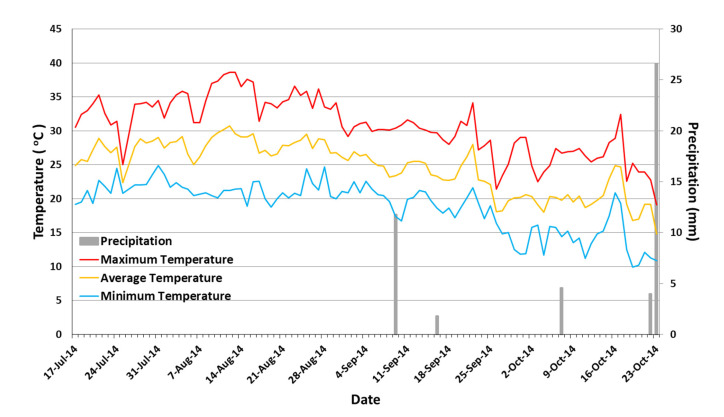
Ambient maximum, minimum, and average temperatures (°C) and precipitation (mm) during the study period (17 July–23 October 2014).

**Figure 2 sensors-23-05802-f002:**
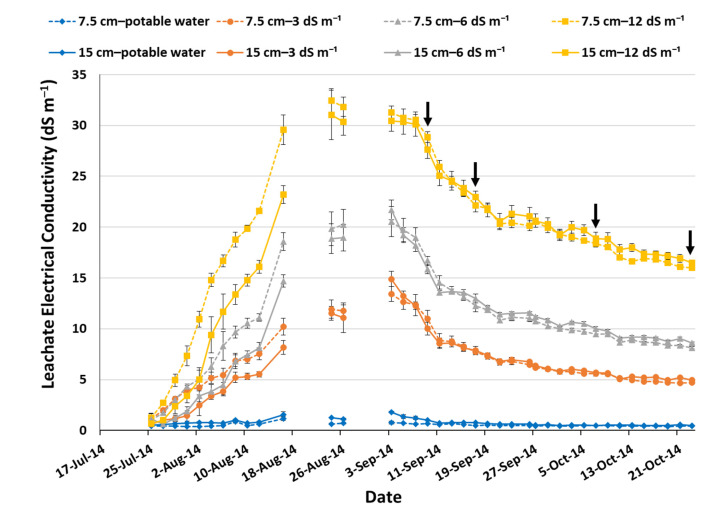
Leachate electrical conductivity (dS m^−1^) during the study period for the substrate depths of 7.5 cm and 15 cm when irrigation was applied with water of electrical conductivity 0.3 dS m^−1^ (potable water), 3 dS m^−1^, 6 dS m^−1^, and 12 dS m^−1^. Error bars represent ± standard error. Arrows indicate rainfall events.

**Figure 3 sensors-23-05802-f003:**
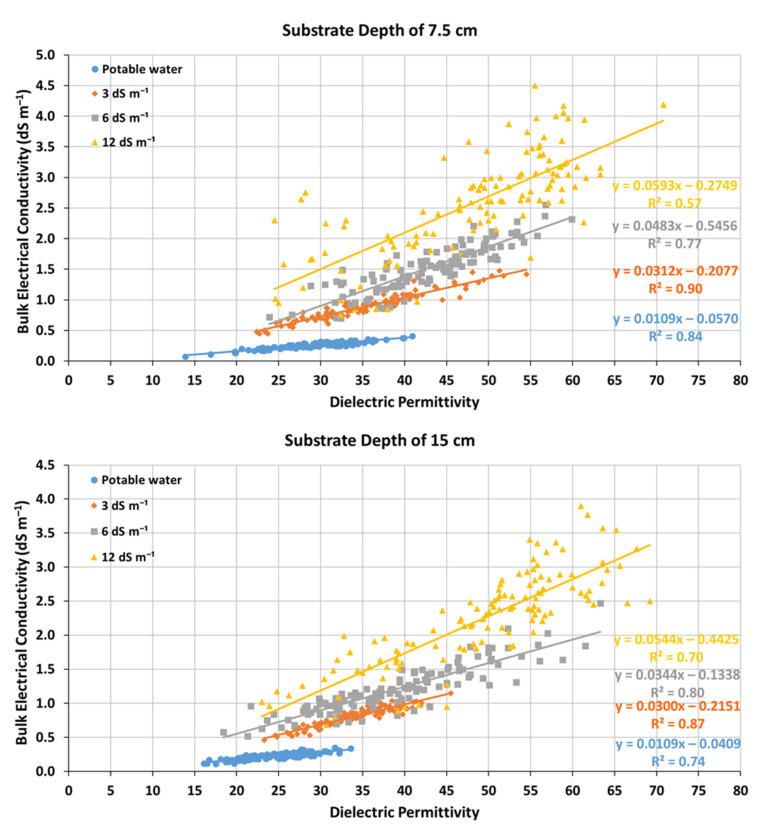
Relationship between bulk electrical conductivity (σ_b_, dS m^−1^) and substrate dielectric permittivity (ε_b_) as determined using the WET-2 dielectric sensor during the study period for the substrate depths of 7.5 cm and 15 cm when irrigation was applied with water of electrical conductivity 0.3 dS m^−1^ (potable water), 3 dS m^−1^, 6 dS m^−1^, and 12 dS m^−1^.

**Figure 4 sensors-23-05802-f004:**
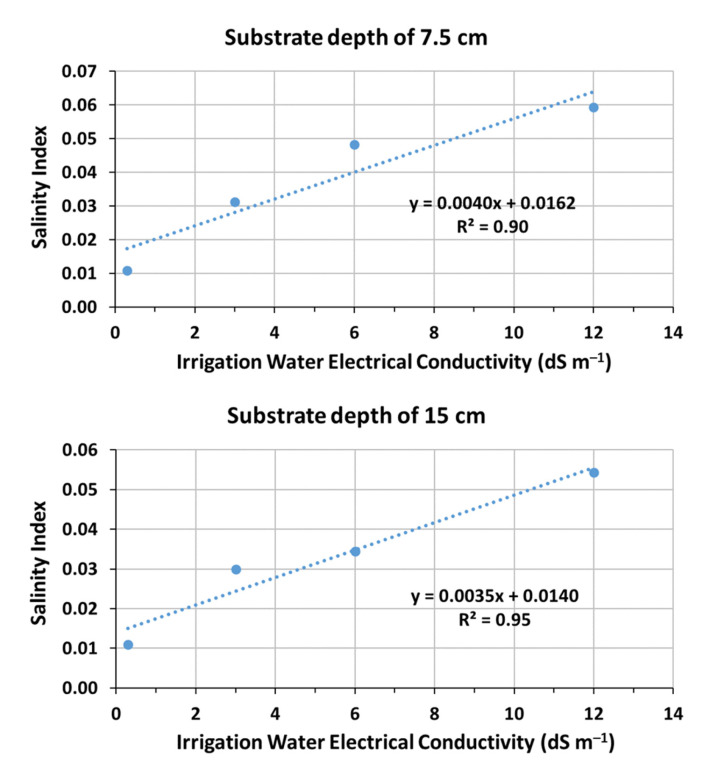
Relationship between salinity index (X_s_) and irrigation water electrical conductivity (EC_i_) of 0.3 dS m^−1^ (potable water), 3 dS m^−1^, 6 dS m^−1^, and 12 dS m^∓1^ for the substrate depths of 7.5 cm and 15 cm.

**Table 1 sensors-23-05802-t001:** Physical and chemical properties of the substrate, which comprised 65% pumice, 15% thermally treated attapulgite clay, 15% grape marc compost, and 5% clinoptilolite zeolite by volume. Values represent the mean values of three replications whenever the standard error is provided (±SE).

Parameter	Value (±SE)	Mechanical Analysis	
		Particle Size	Percent Retained
		mm	% (*w*/*w*)
pH (CaCl_2_)	7.2 (±0.02)	9.5–6.3	1.9
Electrical conductivity, dS m^−1^ (water, 1:10, *m*:*v*)	0.60 (±0.02)	6.3–3.2	23.6
Dry bulk density, kg L^−1^	0.80 (±0.02)	3.2–2.0	17.3
Saturated bulk density, kg L^−1^	1.30 (±0.05)	2.0–1.0	25.9
Bulk density at maximum field capacity, kg L^−1^	1.20 (±0.03)	1.0–0.25	20.4
Maximum water holding capacity, % (*v*/*v*)	54.2 (±1.65)	0.25–0.05	4.4
Total pore volume, %	63.8 (±2.30)	0.05–0.002	5.4
Hydraulic conductivity, mm·min^−1^	7.62 (±0.67)	<0.002	1.1
Plant available water, % *(v*/*v)*	7.8 (±0.30)		

**Table 2 sensors-23-05802-t002:** Prediction of mean substrate electrical conductivity using the salinity index method.

Irrigation Water Electrical Conductivity (dS m^−1^)	Salinity Index	Regression Slope between Salinity Index and Irrigation Water Electrical Conductivity	Estimated Mean Substrate Electrical Conductivity (dS m^−1^)	Measured Mean Leachate Electrical Conductivity (dS m^−1^)	Relative Error (%)
	Substrate depth of 7.5 cm				
0.3	0.0109	0.0040	2.73	0.53	415.1
3	0.0312	0.0040	7.80	6.86	13.7
6	0.0483	0.0040	12.08	11.14	8.4
12	0.0593	0.0040	14.83	19.73	−24.8
	Substrate depth of 15 cm				
0.3	0.0109	0.0035	3.11	0.78	298.7
3	0.0300	0.0035	8.57	7.51	14.1
6	0.0344	0.0035	9.83	10.54	−6.7
12	0.0544	0.0035	15.54	18.56	−16.3

## Data Availability

Not applicable.

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
