# Peer review of "Use of a Dielectric Sensor for Salinity Determination on an Extensive Green Roof Substrate"

_sensors, 2023, doi:10.3390/s23135802_

Round 1
Reviewer 1 Report
In my view, the experiment is justified and conducted well, and the conclusions are sound. As a result, we can learn something new from this paper about the control of salinity in green roof systems. So, the manuscript is acceptable for publication in this Journal. In the below lines I make some comments and suggestions that will help to improve the manuscript.
L14: The authors use ECs as abbreviation of substrate EC in the paper. Since I understand that ECs = pore water EC, it would be convenient to include this clarification in brackets: ECs (pore water EC).
L21: “σb, and bulk dielectric permittivity, εs, of the substrate…” In order to be consistent with the abbreviations adopted in the text (and other papers), I would say “εb” instead of “εs”
L70: “the dielectric permittivity (εs), the bulk electrical...” Include bulk before dielectric.
L102-103: Was it verified that the WET-2 probe measures well in coarse-textured substrates? A faulty contact between the probe's needles and the substrate could impact the accuracy of readings obtained in coarse-textured substrates.
L160: “was irrigated every other day at the depth of 14 mm…” How was the irrigation criteria chosen?
L193-198: Please, indicate that the WET-2 probe is not able to automatically record data.
Figure 1. I suggest showing the mean vapor pressure deficit (VPD) instead of temperature. VPD is a more accurate indicator on the evapotranspiration demand and, therefore, you can better explain changes in leaching fraction (see L220-221)
Table 1: If possible, available water to plants (water retained between -1 and -10 kPa) should be included. Although it is not necessary to achieve the objectives, it would help to understand the irrigation management in this substrate.
M&M: The experiment studies two factors: salinity and substrate depth. Hence, a two-way ANOVA should be used to data statistical analysis. But according to objectives of the work, a regression study of the data is the most proper statistical analysis. The authors should include a brief paragraph discussing how they analyzed the statistical relationships between the experimental variables, including the software used for analysis.
R&D: Paspalum vaginatum. Can the authors include any comments on how growth or emergence was affected by high salinity, if they were affected?
L315-318: The authors considered the leachate EC as the real salinity of the substrate. I think that should be further justified.
Reviewer 2 Report
Hello,
The article is interesting, experimental, relevant.
A few drawbacks:
The text must contain references to formulas (formula no.).
The scheme of the research model should be requested in the methodology.
The methodology must specify the statistical data processing model, and describe which indicators are used for which calculations.
A separate discussion is missing. Too little scientific literature is reviewed and cited in the discussion. A total of 33 literature sources were reviewed. Of these, 29 are reviewed in the Introduction section. Another one (source 30) is cited in the methodology section. So there are only three new sources left for the discussion.
Round 2
Reviewer 2 Report
Hello,
since the article has been improved according to my comments - I think it can be published
Author Response
We would like to extend our sincere gratitude to the reviewer for recommending the publication of our manuscript. We are truly grateful for the valuable feedback and support throughout the review process.